# Skeletal Phenotypes Due to Abnormalities in Mitochondrial Protein Homeostasis and Import

**DOI:** 10.3390/ijms21218327

**Published:** 2020-11-06

**Authors:** Tian Zhao, Caitlin Goedhart, Gerald Pfeffer, Steven C Greenway, Matthew Lines, Aneal Khan, A Micheil Innes, Timothy E Shutt

**Affiliations:** 1Departments of Medical Genetics and Biochemistry & Molecular Biology, Cumming School of Medicine, Alberta Children’s Hospital Research Institute, Hotchkiss Brain Institute, University of Calgary, Calgary, AB T2N 4N1, Canada; tian.zhao@ucalgary.ca; 2Departments of Pediatrics and Medical Genetics, Cumming School of Medicine, Alberta Children’s Hospital Research Institute, Hotchkiss Brain Institute, University of Calgary, Calgary, AB T2N 4N1, Canada; Caitlin.Goedhart@albertahealthservices.ca (C.G.); Matthew.Lines@albertahealthservices.ca (M.L.); Micheil.Innes@albertahealthservices.ca (A.M.I); 3Departments of Clinical Neurosciences and Medical Genetics, Cumming School of Medicine, Hotchkiss Brain Institute, Alberta Child Health Research Institute, University of Calgary, Calgary, AB T2N 4N1, Canada; gerald.pfeffer@ucalgary.ca; 4Departments of Pediatrics, Cardiac Sciences and Biochemistry & Molecular Biology, Cumming School of Medicine, Alberta Children’s Hospital Research Institute and Libin Cardiovascular Institute, University of Calgary, Calgary, AB T2N 4N1, Canada; steven.greenway@albertahealthservices.ca; 5Departments of Pediatrics and Medical Genetics, Cumming School of Medicine, Alberta Children’s Hospital Research Institute, University of Calgary, Calgary, AB T3B 6A8, Canada; Aneal.Khan@albertahealthservices.ca

**Keywords:** mitochondrial disease, skeletal abnormality, protein homeostasis, protein import

## Abstract

Mitochondrial disease represents a collection of rare genetic disorders caused by mitochondrial dysfunction. These disorders can be quite complex and heterogeneous, and it is recognized that mitochondrial disease can affect any tissue at any age. The reasons for this variability are not well understood. In this review, we develop and expand a subset of mitochondrial diseases including predominantly skeletal phenotypes. Understanding how impairment ofdiverse mitochondrial functions leads to a skeletal phenotype will help diagnose and treat patients with mitochondrial disease and provide additional insight into the growing list of human pathologies associated with mitochondrial dysfunction. The underlying disease genes encode factors involved in various aspects of mitochondrial protein homeostasis, including proteases and chaperones, mitochondrial protein import machinery, mediators of inner mitochondrial membrane lipid homeostasis, and aminoacylation of mitochondrial tRNAs required for translation. We further discuss a complex of frequently associated phenotypes (short stature, cataracts, and cardiomyopathy) potentially explained by alterations to steroidogenesis, a process regulated by mitochondria. Together, these observations provide novel insight into the consequences of impaired mitochondrial protein homeostasis.

## 1. Introduction

Mitochondrial disease is a heterogeneous collection of disorders resulting from primary mitochondrial dysfunction due to mutations in either the nuclear genome or the mitochondrial genome (mtDNA (mitochondrial DNA)) [1,2]. Historically, mitochondrial disorders are associated with defects in oxidative phosphorylation and ATP production, with clinical manifestations that tend to be progressive, multisystem, and often affect energy-demanding tissues such as the brain, nerve, and muscle. Moreover, due to the well-recognized role of mitochondria in energy production, it has long been the case that many or most mitochondrial disorders are conceived of as primary disorders of cellular energy production. However, the advent of next generation sequencing has revolutionized how we interpret the characteristics of a mitochondrial disease and has led to the identification of many novel pathogenic variants in mitochondrial proteins [2,3,4], many of which do not necessarily lead to classical clinical phenotypes of mitochondrial disease [3,4,5,6]. For example, disorders of the bones and joints have recently been characterized as one of the uncommon phenotypes of patients presenting with mitochondrial disease [7]. Thus, our understanding of mitochondrial dysfunction in human disease continues to evolve. For the purpose of this review, we define mitochondrial disease as (i) genetic diseases in which the affected gene product primarily functions in the mitochondrion, and (ii) excludes disorders primarily belonging to another well-defined category of inborn error (e.g., fatty-acid oxidation, urea cycle, organic acids, and amino acids catabolism).

Although abnormal cellular bioenergetics are certainly evident in many mitochondrial diseases, energy deficiency itself cannot fully account for the marked heterogeneity of mitochondrial disease phenotypes [5]. Notably, mitochondria have many other important cellular roles, impairment of which can also contribute to disease. Some of these other mitochondrial functions include: mediating apoptosis, calcium signaling, regulating innate immunity, and participation in a variety of metabolic pathways such as steroidogenesis and nucleotide synthesis. Exactly how impairment of these other mitochondrial functions impacts human pathology and contributes to mitochondrial disease remains largely unresolved.

While clinical exome sequencing has increasingly permitted the specific diagnosis and accurate genetic stratification of mitochondrial diseases, an as-yet unrealized aim of mitochondrial medicine is a precise mapping of individual mitochondrial functions onto individual clinical phenotypes. One way to gain mechanistic insight into how impairment of certain functions leads to specific phenotypes is through genetic screens. Historically, researchers have used genetic screens in model organisms to link phenotype to function. Such screens have been critical for understanding many basic biological functions. A key example in mitochondrial research is using the baker’s yeast *S. cerevisiae* to identify genes that cause a “petite” phenotype of small colonies due to lack of mitochondrial oxidative phosphorylation [6,7]. While genetic screens have proven invaluable in the past, in the post-exome era, patients themselves are increasingly the model system in which novel disease genes are being discovered.

Here, we will examine a specific subset of diseases caused by pathogenic variants in mitochondrial proteins with interesting skeletal phenotypes, including skeletal dysplasia, skeletal malformations, and metabolic bone disease [8]. Intriguingly, many of the conditions discussed here are causally related to factors involved in mitochondrial protein homeostasis, specifically mitochondrial protein import.

### 1.1. Mitochondrial Protein Homeostasis 

Part of the phenotypic complexity that characterizes mitochondrial disorders arises from the mitochondrion’s proteomic complexity. While the mtDNA encodes thirteen proteins that are translated by a dedicated mitochondrial translation machinery, the remaining ~1500 nuclear-encoded mitochondrial proteins are translated in the cytosol and imported into mitochondria. Nuclear mitochondrial protein expression and import must be coordinated with mtDNA gene expression, in particular for macromolecular complexes such as mitochondrial ribosomes and oxidative phosphorylation subunits (i.e., complexes I, III, IV, and V), which are comprised of subunits encoded by both nuclear and mitochondrial genomes. In addition to import and translation, proper folding and protein degradation contribute to maintaining mitochondrial protein homeostasis. As such, mitochondria have a set of chaperones to ensure proper folding, as well as a dedicated set of proteases that can degrade damaged or unwanted protein [9]. Notably, many of these chaperones and proteases are also involved in mitochondrial protein import.

Impaired mitochondrial protein homeostasis can lead to activation of the so-called mitochondrial unfolded protein response (UPRmt). Though first described in mammalian cells [10,11], the UPRmt was characterized initially in *C. elegans* [12]. Upon the accumulation of unfolded proteins in the matrix, protein import is reduced. Subsequently, the ATFS-1 transcription factor, which is normally imported into mitochondria for degradation, can go to the nucleus and upregulate a spate of genes, including various mitochondrial chaperones and proteases, to deal with the initial stress. An analogous situation appears conserved in mammalian cells, though it is still not fully understood, as the transcriptional response appears to be part of a larger cellular integrated stress response (ISR) that responds to a variety of cellular and mitochondrial stresses [13]. In mammalian cells, ATF5, a transcription factor that is a functional ortholog of *C. elegans* ATFS-1, is also reported to regulate the expression of mitochondrial protein homeostasis machinery [8]. Notably, a fraction of ATF5 is reported to localize to mitochondria, suggesting a similar organelle partitioning mechanism as ATFS-1 in *C. elegans*. Additionally, other transcription factors such as ATF4 and CHOP are also implicated in mediating the mammalian UPRmt/ISR [9,10]. With respect to disease, the UPRmt/ISR is activated under conditions of cardiac stress [11], and pharmacological induction of the UPRmt/ISR requires ATF5 and is cardioprotective [12]. Nonetheless, how the UPRmt/ISR is activated by and impacts mitochondrial diseases remains to be fully elucidated. 

### 1.2. Mitochondrial Protein Import 

Several interrelated mitochondrial import pathways target proteins to the outer mitochondrial membrane (OMM), the intermembrane space (IMS), the inner mitochondrial membrane (IMM), and the matrix [14] (Figure 1). Generally, mitochondrial protein import is initiated via the TOM (translocase of outer membrane) complex. From the TOM complex, proteins can be inserted into the OMM, with single-spanning and multi-spanning OMM proteins entering through the MIM (mitochondrial import machinery) complex, and β-barrel proteins assembled via the SAM (sorting and assembly machinery) complex. Proteins destined for the IMS can be refolded via a disulfide relay system in the IMS known as the mitochondrial import and assembly (MIA) pathway that mediates proper folding of cysteine-rich proteins. Meanwhile, following transfer through the TOM complex, insertion into the IMM and the matrix occurs via the TIM (translocase of inner membrane) complexes. The TIM23 complex recognizes proteins with a mitochondrial targeting sequence and can import proteins to the matrix, IMM, or IMS. Finally, the TIM22 complex imports multi-spanning proteins into the IMM.

More than half of the nucleus-encoded mitochondrial proteins are targeted to mitochondria via an N-terminal mitochondrial targeting sequence (MTS) composed of an amphipathic alpha-helix. Mitochondria have a set of proteases that mediate the cleavage and removal of the MTS, further linking mitochondrial proteases and import. These proteases are important for maintaining protein homeostasis and mediating mitochondrial protein import. In the matrix, the mitochondrial processing peptidase (MPP) performs the initial cleavage of the MTS [15]. The MPP is composed of two protein subunits. The non-catalytic MPPα subunit, encoded by *PMPCA*, is involved in substrate recognition and binding. Meanwhile, the MPPβ subunit, encoded by *PMPCB*, comprises the catalytic subunit. Critically, MPP cleavage can result in unstable intermediate proteins due to destabilizing N-terminal amino acids according to the N-end rule. To counteract this instability issue, mitochondria harbor two additional distinct proteases that can mediate additional processing events. The first protease is the MIP (mitochondrial intermediate presequence) protease, which is encoded by *MIPEP*. MIP, also known as Oct1, removes an additional octapeptide after MPP cleavage. The second enzyme is an X-pro-aminopeptidase, encoded by *XPNPEP3*, which removes a single N-terminal amino acid. Approximately ~25% of the imported mitochondrial proteins that are cleaved by MPP undergo a secondary cleavage. Notably, cleavage events mediated by both MIPEP and XPNPEP3 are implicated in the activation of CLPP, a subunit of the mitochondrial CLPXP protease that contributes to proper mitochondrial protein homeostasis [16,17]. In the IMM, the inner membrane peptidase (IMP) complex cleaves the MTS from proteins that are partially inserted into the IMM, mediating their release into the IMS. The IMP complex is made up of two proteolytically active subunits, IMP1 and IMP2, encoded by *IMMP1L* and *IMMP2L*, respectively [18].

## 2. Skeletal Phenotypes due to Disorders of Mitochondrial Protein Function

The recognition of skeletal features in patients with pathogenic mutations in *LONP1* and *HSPA9* led to an initial grouping of these syndromes as ‘mitochondrial chaperonopathies’ [19,20,21], as these genes encode a protease and chaperone, respectively. However, as discussed below, pathogenic variants causing skeletal phenotypes have also been found in other mitochondrial proteins that do not encode chaperones (Table 1). Moreover, there are pathogenic variants in other mitochondrial chaperones and proteases that do not cause overt skeletal phenotypes (reviewed in Table 2). Thus, the term mitochondrial chaperonopathy does not fit for some of these skeletal disorders. Instead, we refer to these disorders here as ‘mitochondrial skeletal disorders’.

### 2.1. LONP1 and CODAS (Cerebral, Ocular, Dental, Auricular, and Skeletal Syndrome)

*LONP1* encodes a multi-functional member of the AAA+ superfamily of ATPases, which is localized to the mitochondrial matrix where it has protease, chaperone, and mtDNA-binding activities [22,23]. In addition to maintaining general protein homeostasis, LONP1 regulates mtDNA stability and expression via processing of the mitochondrial transcription factors TFAM [19] and MTERF3 [24]. Notably, LONP1 physically associates with MPP, potentially accounting for the role LONP1 plays in processing newly imported mitochondrial proteins [24]. Although constitutively expressed, LONP1 is upregulated during acute cellular stresses including heat shock and oxidative damage, and is a key mediator of the UPRmt/ISR [24].

CODAS was initially described in 1991 as an autosomal recessive disorder encompassing a distinctive set of phenotypes including spondyloepiphyseal dysplasia, short stature, dental anomalies, cataracts, hearing loss, developmental delay, and variable dysmorphism. [25]. However, the identification of *LONP1* as the causative gene was not until 2015 [19,20]. The identification of a mitochondrial gene was unexpected for a developmental dysmorphology disorder, as severe skeletal phenotypes (i.e., spondyloepiphyseal dysplasia) had not been recognized previously in mitochondrial diseases. As with many mitochondrial diseases, the phenotype of patients with LONP1 pathogenic variants varies widely. Since the initial identification of *LONP1* as the causative gene for CODAS, reports have emerged with additional *LONP1* variants associated with an expanding set of clinical phenotypes [26,27,28,29,30]. These presentations vary from milder presentations that include only cataracts with minimal extraocular findings [28], to an atypical presentation more similar to Marinesco-Sjogren syndrome [26], and even classic mitochondrial disease phenotypes associated with mtDNA depletion [27,29,30]. While LONP1 pathogenic variants are generally recessive, a heterozygous gain-of-function variant exhibiting excessive proteolytic activity and loss of chaperone function was recently proposed to be the cause of a neonatal-onset disorder characterized by epilepsy, pachygyria, diffuse progressive white matter atrophy, global developmental delay, and death in infancy [31].

This phenotypic variability raises an interesting question about how LONP1 dysfunction leads to disease. Described *LONP1* variants causing CODAS in humans are generally substitutions or small, in-frame deletions in the conserved ATP-binding and proteolytic domains of LONP1 [26,30]. Moreover, *LONP1* knockout mice exhibit embryonic lethality [22]. Thus, it is likely that this protein is essential, which suggests that patients with *LONP1* variants must retain some LONP1 activity. Therefore, it is possible that the severity of phenotypes associated with *LONP1* variants may be linked to the severity of the dysfunction imparted by the sequence changes and/or impairments in substrate recognition caused by the various *LONP1* variants. Further supporting this notion is the observation that the R721G and P676S variants causing CODAS do not affect TFAM degradation in an in vitro assay, but do affect the degradation of another LONP1 target, STARD1 [19]. Moreover, basal mitochondrial respiration in patient fibroblasts with the R721G variant causing classic CODAS is largely unaffected, with only a slight decrease in spare respiratory capacity, suggesting that a major defect in oxidative phosphorylation may not be the driving force of pathogenicity for CODAS. Meanwhile, the Y565H variant linked to mtDNA depletion reduces the ability of LONP1 to bind and degrade TFAM [27], a master regulator of mtDNA. This difference in substrate recognition may explain the mtDNA depletion and more severe subsequent mitochondrial dysfunction in oxidative phosphorylation. 

### 2.2. HSPA9 and EVEN-PLUS (Epiphyseal, Vertebral, and Ocular Changes Plus Associated Findings of Severe Microtia, Nasal Hypoplasia, and Other Malformations)

*HSPA9* encodes heat shock protein A9 (HSPA9/mtHsp70/Mortalin/Grp75), a chaperone that functions in folding mitochondrial matrix proteins, but has also been attributed to peroxisomal regulation [32], amongst other cellular localizations and functions [33]. HSPA9 interacts with hydrophobic portions of unfolded proteins to prevent protein aggregation, and associates with the TIM23 import machinery to assist in mitochondrial import into the matrix. [34,35]. Notably, HSPA9 interacts with LONP1 and is important for mediating its proteolytic activity [21,34], and is also upregulated by the UPRmt/ISR [36].

Recessive compound heterozygous or homozygous variants in *HSPA9* cause EVEN-PLUS syndrome, which is characterized by skeletal phenotypes resembling CODAS, with additional findings [21]. The two *HSPA9* variants initially linked to EVEN-PLUS syndrome are found in the nucleotide binding domain of HSPA9. Biochemically, both variants reduce ATP hydrolysis required for chaperone activity of the protein and decrease the stability of the protein [37]. More recently, an additional EVEN-PLUS patient with novel variants in *HSPA9* has been identified, adding to the clinical variability of the disorder [38]. Meanwhile, variants in *HSPA9* have also been associated with sideroblastic anemia from recessive (or pseudo-dominant) mutations [39] and are potentially associated with Parkinson disease [40].

## 3. Expanding the List of Mitochondrial Skeletal Disorders

In addition to *LONP1* and *HSPA9*, a few other disease genes encoding mitochondrial proteins have been linked to genetic skeletal disorders, where they have been classified as spondyloepimetaphyseal or severe spondysplastic dysplasias [8] (Table 1). Around the same time that *LONP1* and *HSPA9* were identified as skeletal disorders, pathogenic variants in the mitochondrial tRNA synthetase *IARS2* [41] and the mitochondrial import chaperone *PAM16* [42] were also linked to skeletal phenotypes. Meanwhile, pathogenic variants in *AIFM1* were recognized to cause skeletal phenotypes resembling CODAS and EVEN-PLUS [43]. More recently, pathogenic variants in *PISD*, encoding an enzyme involved in mitochondrial lipid synthesis, were also linked to skeletal defects [44,45,46]. The fact that none of these genes encode a chaperone suggests that additional mitochondrial functions may also contribute to the development of skeletal abnormalities.

### 3.1. IARS2 and CAGSSS (Cataracts, Growth Hormone Deficiency, Sensory Neuropathy, Sensorineural Hearing Loss, Skeletal Dysplasia Syndrome)

*IARS2* encodes the mitochondrial isoleucyl-tRNA synthetase, an enzyme that attaches the isoleucyl amino acid to its cognate tRNA for translation of mitochondrial-encoded proteins. The first report of CAGSSS, an autosomal recessive syndrome with cataracts, short-stature secondary to growth hormone deficiency, hearing loss, peripheral neuropathy, and skeletal dysplasia, was described in three patients with variants in *IARS2* [41]. As skeletal dysplasias had not previously been described in association with mitochondrial dysfunction, the initial description of skeletal features was questioned as possibly secondary, or due to a concurrent disorder [47]. However, as discussed above with CODAS and EVEN-PLUS, and the growing list of mitochondrial diseases presented in this review, skeletal features are gaining traction as a feature of mitochondrial disease. Moreover, it is now clear that pathogenic variants in *IARS2* are indeed causative and directly linked to the skeletal dysplasia phenotype, as multiple examples of homozygous and compound heterozygous missense mutations have been described [48,49,50,51,52]. Additional features of note described in these patients include neurodevelopmental delay and seizures. Interestingly, the majority of reported mutations in *IARS2* are either intronic or missense. Hence, it is likely that IARS2 is essential for human functioning and complete loss of its function is lethal. Somewhat surprisingly, no obvious defects in mitochondrial translation were evident in patient fibroblasts harboring *IARS2* variants [41].

### 3.2. PAM16 and SMDMDM (Spondylometaphyseal Dysplasia, Megarbane–Dagher–Melki Type)

PAM16 (presequence translocase-associated motor 16), also known as MAGMAS (mitochondria-associated granulocyte macrophage colony-stimulating factor-signaling), is a critical mediator of mitochondrial protein import into the mitochondrial matrix via the TIM23 complex [53,54]. PAM16 functions in a complex with the J-type domains of DNAJ homologs, which stimulate the ATPase activity of HSPA9 [55,56]. Thus, in addition to a clear functional link to HSPA9, PAM16 is primarily involved in mitochondrial protein import.

The Megarbane–Dagher–Melki type of spondylometaphyseal dysplasia (SMDMDM) was first described in two siblings with chondrodysplasia, short stature, developmental delay, and cardiomyopathy who died before the age of two [57]. Later, a second family with similar phenotypes was described [58], and a homozygous missense variant in *PAM16* (N76D) was identified in both families [42]. More recently, a slightly milder phenotype of SMDMDM was also reported in another patient harboring a nearby variant in *PAM16* (Q74P) [59].

### 3.3. AIFM1 and SEMD-HL (Spondyloepimetaphyseal Dysplasia, X-linked, with Hypomyelinating Leukodystrophy)

Apoptosis-inducing factor (*AIFM1*) encodes a protein first identified for its role in promoting apoptosis when released from the mitochondrial IMS [60]. However, AIFM1 also has other distinct functions in the mitochondria. For example, AIFM1 has a FAD-dependent NADH-oxidase activity that functions in the assembly of mitochondrial complexes I and III [61,62,63]. In addition, AIFM1 contains an HSP70 binding domain [64]. Similar to HSPA9 and LONP1, AIFM1 is also involved in protein import, as it interacts functionally with GFER and affects the stability of MIA40 [65,66,67], two key proteins that mediate protein import to IMS via the MIA pathway.

Recently, SEMD-HL was reported as another mitochondrial disorder linking skeletal phenotypes with mitochondrial dysfunction resembling CODAS syndrome and EVEN-PLUS [43,68]. SEMD-HL is characterized by spondyloepimetaphyseal dysplasia, midface hypoplasia, and hypomyelinating leukodystrophy with progressive neurodegeneration of the central and peripheral nervous system. The X-linked recessive *AIFM1* variant causing SEMD-HL occurs in the middle of AIFM1′s FAD-binding domain [68].

In addition to SEMD-HL, pathogenic *AIFM1* variants have been associated with other syndromes [69]. These diseases include: Cowchock syndrome (X-linked Charcot–Marie–Tooth disease, CMTX4) characterized by axonal sensorimotor neuropathy, deafness, and cognitive impairment [62,70,71], X-linked recessive sensorineural hearing loss with sensory polyneuropathy [72] (also described in milder form as X-linked dominant [73]), cerebellar ataxia [74,75], and mitochondrial encephalomyopathy, ranging from mild [76,77] to severe phenotypes [63,78,79,80,81]. These different syndromes associated with distinct variants are reminiscent of the situation with *LONP1* and *HSPA9*, suggesting that the different variants either have different severity or affect different functions/substrates of AIFM1 [62].

### 3.4. PISD

The *PISD* gene encodes an enzyme that localizes to the IMM, where it converts phosphatidylserine to phosphatidylethanolamine, a critical lipid that is important for mitochondrial function [82,83,84]. Rather than exerting a direct effect on mitochondrial protein homeostasis, impairment of PISD activity affects mitochondrial proteases embedded in the IMM (i.e., OMA1, YME1L) by altering the lipid composition of this critical membrane [45,85]. 

Recently, a number of different groups identified novel mutations in the mitochondrial enzyme PISD, in patients with a variety of skeletal phenotypes [44,45,46]. Zhao et al. described two sisters with short stature, mid-face hypoplasia, congenital cataracts, and white matter changes with compound heterozygous variants (one missense, Arg277Gln, and one splice variant) [45]. Meanwhile, Girisha et al. described unrelated patients with spondyloepimetaphyseal dysplasia and a different homozygous substitution [44]. Both reported that substitutions impair an autocatalytic processing event required to generate the catalytic site of the PISD enzyme, and the Arg277Gln variant was shown to impair the activity of IMM proteases [45]. Finally, Peter et al. described patients harboring a splicing variant at the 3′ end of the PISD gene with early-onset retinal degeneration, sensorineural hearing loss, microcephaly, intellectual disability, and skeletal dysplasia with scoliosis and short stature resembling Liberfarb syndrome [46]. In all cases, it is expected that patients maintain some residual levels of PISD activity, as complete loss of PISD is embryonic lethal in mice [82], and thus far no clinical patient has been reported with biallelic truncating variants. Reminiscent of LONP1 variants causing CODAS, the milder end of the *PISD* phenotypic spectrum presents essentially as a multiple malformation syndrome lacking the classical clinical characteristics of a mitochondrial disorder.

## 4. Integrating the Mitochondrial Functions Underlying Skeletal Phenotypes

Many of the conditions discussed above are caused by pathogenic variants in genes directly involved in mitochondrial protein homeostasis, including mitochondrial chaperones, proteases, or protein import factors. However, it is less obvious how IMM lipid homeostasis (PISD) and mitochondrial tRNA aminoacylation (IARS2) might also link to mitochondrial protein homeostasis or mitochondrial import. Understanding how these processes may be connected could provide insight into the mechanistic underpinnings of the skeletal phenotypes in mitochondrial disease patients. Thus, we also briefly examine additional mitochondrial diseases associated with these functions.

### 4.1. Linking Mitochondrial Protein Import to Mitochondrial Proteases and Chaperones

As alluded to above, there is a reciprocal relationship between mitochondrial proteostasis and mitochondrial protein import. After being translated in the cytosol, nuclear-encoded mitochondrial proteins are imported into mitochondria in an unfolded state where they become substrates for mitochondrial proteases and chaperones that aid in protein import. Meanwhile, impaired mitochondrial protein homeostasis can cause reduced protein import, which is thought to be a key step in activating the UPRmt/ISR that upregulates several mitochondrial proteases and chaperones. In the context of discussing the role of mitochondrial import in mitochondrial skeletal disorders, it is also worth noting that pathogenic variants in other proteins directly involved in mitochondrial protein import have been described [86]. However, it has not been determined whether the UPRmt/ISR is activated in mitochondrial skeletal disorders or these mitochondrial protein import disorders, nor what impact the UPRmt/ISR might have on their pathogenesis. The specific disease genes linked to mitochondrial protein import and general patient phenotypes are described in Table 3. As discussed in more detail later, many of these diseases share phenotypes that can also be found in affected individuals with mitochondrial skeletal disorders.

### 4.2. Mitochondrial Lipid Homeostasis

Lipids are critical for the proper function of mitochondria, which comprise two distinct membranes. The OMM mediates interactions between mitochondria, the cytosol, and other organelles. The IMM consists of several functionally specialized domains, including invaginations (cristae), which house the electron transport chain complexes, and maintain the electrochemical gradient required to generate ATP. In order to maintain these functions, the IMM is rich in two key lipids that support membrane curvature, phosphatidylethanolamine and the mitochondrial-specific lipid cardiolipin. In addition to *PISD*, there are a few other disease genes linked to mitochondrial lipid homeostasis (Table 4). These genes include *TAZ*, *AGK*, and *DNAJC19*, mutations in which cause Barth syndrome [87,88], Senger syndrome [89,90], and DCMA (dilated cardiomyopathy with ataxia) syndrome [91], respectively. *TAZ* encodes a cardiolipin transacylase required cardiolipin remodeling [92]. *AGK* encodes acylglycerol kinase, an enzyme that produces phosphatidic and lysophosphatidic acids in the IMM. Meanwhile, DNAJC19 encodes a protein implicated in regulating cardiolipin remodeling [93]. Notably, both DNAJC19 and AGK are further described to play roles in mitochondrial protein import [94,95,96,97].

#### Connecting Mitochondrial Lipids to Mitochondrial Protein Homeostasis and Import

Many mitochondrial proteases are embedded within the IMM [98,99], where they are positioned to sense changes in lipid composition. Importantly, recent work has shown that alterations to the IMM composition of phosphatidylethanolamine affects these proteases and impacts mitochondrial proteins [45,85]. For example, both decreased PISD activity and overexpression of PISD reduce the expression of the IMM protease OMA1 [45], while PISD depletion also impacts the IMM protease YME1L [85]. Given that the mitochondrial protein import machinery also resides within the mitochondrial membranes, it is likely that mitochondrial protein import is also affected by changes in lipid composition [100]. In this regard, the recent identification of the AGK as a component of the TIM22 import complex [95,96], and the role of DNAJC19 in mitochondrial protein import [94], are intriguing. Meanwhile, cardiolipin is an important mediator of the interaction between TIMM50 and the TIM23 complex [101]. However, future work will need to be performed to further explore the functional link between mitochondrial protein import and lipid composition.

### 4.3. Mitochondrial tRNAs

Considering the initial description of pathogenic variants in *IARS2* causing CAGSSS, it is worth examining the role of mitochondrial tRNA maturation in mitochondrial translation. In addition to the ~1500 proteins that are imported into mitochondria, the human mitochondrial genome encodes thirteen proteins that are essential subunits of the electron transport chain and ATP synthase. These thirteen IMM proteins are translated by a dedicated mitochondrial translation machinery, impairment of which can also impact mitochondrial protein homeostasis. The mitochondrial genome also encodes two ribosomal mt-RNAs and 22 mt-tRNAs required for mitochondrial translation. 

Following their transcription from the mitochondrial genome, mt-tRNAs undergo several important processing events and post-transcriptional modifications. First, cleavage at the 5′ and 3′ ends of mt-tRNAs is critical for the processing of the polycystronic mitochondrial transcripts into individual mRNAs, tRNAs, and rRNAs [102]. Next, certain bases within mt-tRNAs undergo post-transcriptional modifications that are thought to help with proper folding and function. Finally, like cytosolic tRNAs, mt-tRNAs require the addition of 3′ CCA trinucleotide to form a functional molecule that can then be aminoacylated. This aminoacylation step, performed by tRNA synthetases specific for each tRNA, adds the cognate amino acid that will be used for translation [103]. Given this complexity in generating proper mt-tRNAs, it should come as no surprise there are many steps that can be impaired, potentially leading to disease [104]. Here, we focus on aminoacyl-tRNA synthetases and their roles in mitochondrial disease.

#### 4.3.1. Mitochondrial Aminoacyl-tRNA Synthetases

A growing list of mitochondrial aminoacyl-tRNA synthetases are now recognized to cause disease with a surprisingly broad variety of human clinical phenotypes, especially given their common role in mitochondrial translation [103,105]. While IARS2 is the only aminoacyl-tRNA synthetase linked to skeletal dysplasia, pathogenic variants in a subset of other mitochondrial aminoacyl-tRNA synthetases cause other phenotypes that overlap with what is seen in patients with mitochondrial skeletal disorders, suggesting there may be a common underlying dysfunction (Table 5). The fact that not all pathogenic variants in mitochondrial aminoacyl tRNA synthetases cause skeletal phenotypes raises the question of why this is so, with one explanation for this variability being that the severity of the dysfunction may vary, and there is a certain threshold that must be met for certain patient phenotypes.

One might assume that dysfunctional aminoacyl-tRNA synthetases would result in a global decrease in mitochondrial translation, and that this would be reflected by a common disease phenotype. However, this does not appear to be the case. Intriguingly, with respect to function, several of the mitochondrial aminoacyl-tRNA synthetase variants do not show obvious evidence of reduced mitochondrial translation, although this has not been investigated for most of the genes and variants reported to date. For example, in *LARS2* patients with less severe presentation, where variants resulted in a ~50% decrease in aminoacylation activity, no changes in steady-state levels of mitochondrial-encoded proteins were observed, suggesting mitochondrial translation was largely unaffected in these cells [106].

The fact that some pathogenic variants in aminoacyl-tRNA synthetases do not cause overt changes in mitochondrial translation suggests that more subtle alterations to mitochondrial function are involved. One possibility is that mitochondrial translation has not often been investigated in the affected tissue (e.g., tissue-specific defects cause the disease). Another possibility is that these proteins may have additional functions, impairment of which could explain the pathology [105]. This latter situation could certainly be the case for *GARS*, which encodes both cytosolic and mitochondrial isoforms. The notion of multiple functions could be extended further to encompass the possibility of moonlighting functions for mitochondrial tRNA synthetases, which may also partially explain the phenotypic variability in patients [103]. In particular, moonlighting functions are well recognized for cytosolic aminoacyl tRNA synthetases [107,108] and have also been seen for the mitochondrial WARS2 aminoacyl tRNA synthetase [109]. However, it is unlikely that multiple mitochondrial aminoacyl tRNA synthetases would acquire the same moonlighting function. Alternatively, it is more likely that there is something else happening mechanistically that we do not currently appreciate fully, and which will require further study to elucidate. 

#### 4.3.2. Connecting Mitochondrial tRNAs to Mitochondrial Protein Homeostasis

We posit that impairment of mitochondrial tRNA function may lead to a proteostatic stress. Although tRNAs are clearly important for mitochondrial translation, global reductions in mitochondrial translation do not appear to be common among the pathogenic variants described above, at least in the cases where this has been investigated. However, it is not unreasonable to assume that more subtle defects in mitochondrial translation, as a result of impaired tRNA aminoacylation, could lead to proteostatic stress, which might add to the burden of mitochondrial proteases and chaperones.

How such subtle defects might influence mitochondrial proteostasis is yet to be determined. However, it is intriguing that several of the tRNA species discussed here are required for the insertion of hydrophobic amino acids into proteins. This fact may be relevant as mitochondrial-encoded proteins are embedded in the IMM, and are thus largely hydrophobic. It is also notable that isoleucine is one of the most abundant amino acids present in mitochondrial-encoded proteins. Perhaps stalled ribosomes lead to unstable partially translated proteins that must be degraded, adding a mitochondrial proteostatic stress. Alternatively, mistranslation events have been demonstrated to induce mitochondrial stress [110], which could also be relevant as another more subtle translation defect. Given the well-established link between mitochondrial protein homeostasis and mitochondrial import, it is not unreasonable to assume that protein import could also be impacted by subtle impairment to mitochondrial translation, as misfolding of IMM proteins can lead to reduced protein import [111].

In addition, there are a number of intriguing links between mitochondrial tRNAs and mitochondrial protein homeostasis. For example, upon treatment with GTPP, there is a reduction in the processing of the long polycystronic RNAs that are cleaved to release tRNAs [112]. GTPP is an inhibitor of the mitochondrial matrix chaperone TRAP1, which induces an acute mitochondrial proteostatic stress that activates the UPRmt/ISR. Critically, this decreased RNA processing appears to be mediated by LONP1-mediated processing of the RNAseP component MRRP3, and is proposed to be part of a response to reduce mitochondrial translation upon proteostatic stress. Thus, mitochondrial proteostasis is linked to tRNA maturation. There are also notable links between aminoacyl tRNA synthetases and protein homeostasis. In DARS2 knockout mice, which have a cardiomyopathy phenotype, deletion of the mitochondrial protease subunit *CLPP* is protective, providing evidence that CLPP plays a role in regulating mitochondrial translation [113]. Moreover, a direct protein interaction between the SARS2 Drosophila homolog SLIMP has been demonstrated to stimulate the proteolytic activity of LONP1 [114]. Finally, it is notable that LONP1 can directly bind to RNA [115], and that a missense variant in LONP1 is associated with altered methylation of the mitochondrial tRNA Gly and tRNA His [116]. Together, these links, combined with the similar phenotypes in patients with impaired mitochondrial protein and tRNA homeostasis, support a functionally relevant interaction between mitochondrial tRNAs and mitochondrial protein homeostasis that will likely continue to be uncovered.

## 5. Additional Features in Mitochondrial Skeletal Disorders

In order to gain additional insight into the underlying cause of mitochondrial skeletal disorders (Table 1), we looked for other common phenotypes that might be shared. At the same time, we also considered many of the other mitochondrial diseases caused by variants in genes that encode proteases and chaperones (Table 2), mitochondrial import proteins (Table 3), regulators of mitochondrial lipids (Table 4), and mitochondrial aminoacyl tRNA synthetases (Table 5). Among the disorders examined, there were several recurring phenotypes that are notable. These phenotypes broadly include neurodevelopmental issues (delayed neurodevelopment, low IQ, epilepsy, autism spectrum disorder, and microcephaly) or movement disorders arising early in life (e.g., ataxia, spastic paraplegia, tremor). In addition, both cataracts and cardiomyopathy (e.g., hypertrophic and dilated) emerged as recurring phenotypes across these disorders. Finally, though none of the additional disorders discussed (Table 2, Table 3, Table 4 and Table 5) here are formally considered to be skeletal disorders, it is notable that they can have phenotypes that are ‘skeletal’ in nature, including short stature/growth retardation, craniofacial abnormalities (e.g., mid-face hypoplasia), or other alterations to the skeletal system (e.g., scoliosis, hip dysplasia, Marfanoid features). Though short stature/growth retardation can be caused by many types of cellular dysfunction (not necessarily impaired bone metabolism), we include it here to bring attention to a phenotypic feature that may not always be appreciated clinically. As such, some patients with this presentation could have milder skeletal dysplasias that are unrecognized. It is important to note that many of these additional features of mitochondrial skeletal disorders are also found in other disorders that are not mitochondrial in nature. Nonetheless, the recurrence of these phenotypes across a variety of mitochondrial diseases linked to mitochondrial protein homeostasis and import, suggests they might share a common mechanistic underpinning with the skeletal dysplasias present in mitochondrial skeletal disorders.

## 6. Discussion

The list of mitochondrial disease genes linked to skeletal abnormalities, which we term mitochondrial skeletal disorders, affects several mitochondrial functions, including chaperones, proteases, protein import, IMM lipid metabolism, and tRNA maturation. To link these mitochondrial functions, we propose a model whereby impaired mitochondrial protein homeostasis leads to reduced mitochondrial protein import, and that impaired import may be the key underlying dysfunction (Figure 2). In these cases, the proteostatic stress could be caused by reduced chaperone capacity, decreased protease activity, or impaired mitochondrial translation.

Although one could imagine a situation where impaired protein import could also cause a proteostatic stress (e.g., partially imported proteins), there is not yet any direct evidence to support this notion. Furthermore, there are several reasons to think that impaired protein import may be necessary and sufficient to mediate the proposed pathway. First, the proteases and chaperones covered here can be linked directly to protein import. Second, though we discussed how altered IMM lipid composition can affect mitochondrial proteases, it is also possible to envision how this aspect of mitochondrial function could also have a direct effect on IMM import. Third, as mitochondrial-encoded proteins are integral membrane proteins that are co-translationally inserted into the IMM, it is reasonable to assume that accumulation of ‘irregular’ proteins would also put a stress on the IMM. Finally, there is accumulating evidence for examples of retrograde signaling that are mediated by impaired mitochondrial protein import, suggesting that functional protein import may act as a rheostat for mitochondrial function.

The most notable example of this type of retrograde signaling is the UPRmt/ISR [13]. Another well-studied example is the Parkinson disease protein PINK1, which promotes mitochondrial autophagy when mitochondrial import is blocked [117], and which can be activated by the accumulation of unfolded proteins in the matrix [118]. In addition, there is a growing appreciation for the interrelationship between mitochondrial and cytosolic protein homeostasis and the importance of mitochondrial protein import [9]. Together, these examples highlight the importance of mitochondrial protein import, as several retrograde pathways can be activated when this process is impaired. While it is expected that such protein import retrograde pathways might be activated in mitochondrial skeletal disorders, this has yet to be confirmed. Moreover, whether constitutive activation of these pathways would protect or aggravate the pathology of these disorders also remains unknown.

In addition to skeletal phenotypes in our list of mitochondrial skeletal disorders, several additional phenotypes are often present, and are shared among other mitochondrial diseases affecting similar mitochondrial functions. This finding is suggestive of a common mechanistic underpinning. Moreover, several of these disorders do not exhibit major defects in oxidative phosphorylation, which are typically associated with mitochondrial disease. Thus, we posit that these shared phenotypes could be due to impairment of another mitochondrial function. 

In this regard, steroidogenesis is a tantalizing candidate as a possible alternative mitochondrial function that could be affected, and it is worth briefly reviewing the role of mitochondria in this process. Pregnenolone, the precursor for all steroid hormones, is made from cholesterol by the mitochondrial enzyme P450ssc [119], which intriguingly interacts with the TIM50 subunit of the TIM23 complex [120]. Notably, import of cholesterol into mitochondria is the rate-limiting step in the production of pregnenolone [121]. Additionally, several other enzymes involved in steroid synthesis are also localized inside mitochondria, including the enzymes that produce glucocorticoids (11β-hydroxylase) [122]. Thus, mitochondria play a critical role in steroidogenesis. 

There are also a few hints suggesting that the production of steroid hormones might be impacted in some of the mitochondrial disorders discussed here. For example, the ovarian failure in Perrault syndrome and the genital abnormalities (e.g., cryptorchidism/hypospadia) associated with variants in *HSPA9* and *DNAJC19* could be due to altered levels of sex hormones. In addition, elevated levels of glucocorticoids could explain several of the other features of mitochondrial skeletal disorders (e.g., short stature/growth retardation, cataracts, and cardiomyopathy), which are shared with other mitochondrial diseases linked to protein import. For example, glucocorticoid therapy is a common cause of faltering growth in children [123], while Cushing syndrome, which is caused by excessive glucocorticoids, can also cause growth retardation [124,125]. Additional common side effects of glucocorticoids include cataracts [126] and cardiomyopathy [127].

Notably, endocrine manifestations have been previously described in mitochondrial disease [128]. However, this link has typically been in the context of more severe phenotypes with impaired oxidative phosphorylation. Given the severe mitochondrial impairment and interrelatedness of many mitochondrial functions, it is likely that multiple mitochondrial functions are impaired in many mitochondrial diseases. Thus, it has not been possible to ascribe these endocrine abnormalities to a specific function. The potential link to steroidogenesis presented here suggests that impaired mitochondrial protein homeostasis/import may be the key, even in the absence of severe mitochondrial dysfunction. However, additional work needs to be done to test this hypothesis and determine the underlying mechanism.

In the context of the skeletal dysplasias described here, while it is clear the impaired mitochondrial protein homeostasis can cause mitochondrial disease, it remains unclear the exact molecular mechanism by which impaired mitochondrial protein homeostasis negatively impacts skeletal development. The expanding role of mitochondrial function in bone, aside from the function of mitochondria in energy production, is not as well understood, as bone is physically more difficult to access compared to tissues like skeletal muscle, has a lower mitochondrial density, and bone tissue pathology does not target examination of mitochondrial structural abnormalities. Interestingly, tetracycline, which has for decades been the standard to look for bone turnover, specifically targets mitochondria in bone and impairs their function [129], and these effects can be manifested in metabolic bone diseases like osteogenesis imperfecta [130]. Notably, estrogen and androgen receptors play critical roles in bone health [131], providing a potential mechanism through which alterations to steroidogenesis could impact bone development. Meanwhile, mitochondrial dysfunction in the PolG mouse model of aging is recognized to impair osteogenesis, suggesting a direct effect of mitochondrial function on osteoclasts and osteoblasts [132]. Finally, while the skeletal impairments in mitochondrial skeletal disorders are rather severe, it is also worth noting that poor bone health is recognized generally in primary mitochondrial disease [133]. 

One of the limitations to our approach and the general description of mitochondrial skeletal disorders is the fact that many of the diseases discussed here are extremely rare, sometimes with only a handful of patients described. The rarity of these disorders makes it difficult to get a complete clinical picture and some of the detailed pathophysiological mechanisms still need to be more clearly understood. However, with the next-generation sequencing approaches quickly gaining traction for the diagnosis of these rare genetic disorders, it is likely that additional patients will be found that harbor new variants in both known and novel disease genes and extend their phenotypic spectra. These discoveries will undoubtedly contribute to a better understanding of mitochondrial functions and how their impairment leads to disease. Meanwhile, the development of animal models will allow a more robust understanding of these diseases, and comparisons amongst them. 

By their nature, rare diseases have been historically under-studied, in part due to the perceived impact of understanding a disease that only affects a small number of people. However, there is a growing appreciation that an understanding of the underlying dysfunction in rare diseases can be applied directly to other human diseases. Thus, lessons taken from the study of mitochondrial diseases may have broad impact on age-related osteopenia and could help to identify future targets for therapy. It will also be interesting to see how the insight gained from the mitochondrial skeletal disorders we describe here could potentially be informative to other disorders linked to mitochondrial dysfunction. For example, mitochondrial quality control and protein import are already implicated in neurodegenerative diseases such as Parkinson disease and Alzheimer disease [134,135].

## Figures and Tables

**Figure 1 ijms-21-08327-f001:**
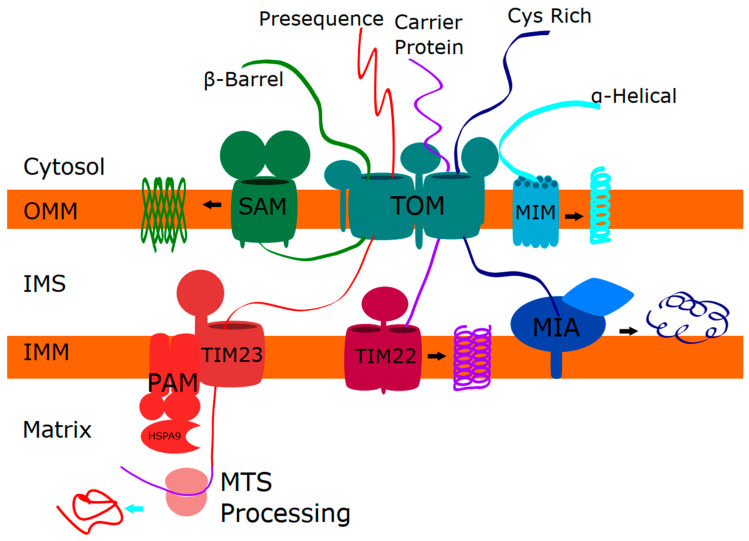
Diagram of mitochondrial protein import pathways. Mitochondrial proteins are imported and targeted to their final destinations through a variety of interrelated pathways starting with the TOM (translocase of outer membrane) complex. Most proteins destined for the OMM (outer mitochondrial membrane) are inserted via the MIM (mitochondrial import machinery) (teal), with β-barrel proteins inserted via the SAM (sorting and assembly machinery) complex (green). Cysteine-rich proteins in the IMS (inner membrane space) are refolded via the MIA (mitochondrial import and assembly) pathway (blue). Proteins containing an MTS (mitochondrial targeting sequence) that are destined for the matrix or IMM (inner mitochondrial membrane) are transferred to the TIM23 complex and the MTS is removed (red). Finally, multi-spanning IMM proteins, such as the SLC family of carrier proteins, are inserted via the TIM22 complex (purple).

**Figure 2 ijms-21-08327-f002:**
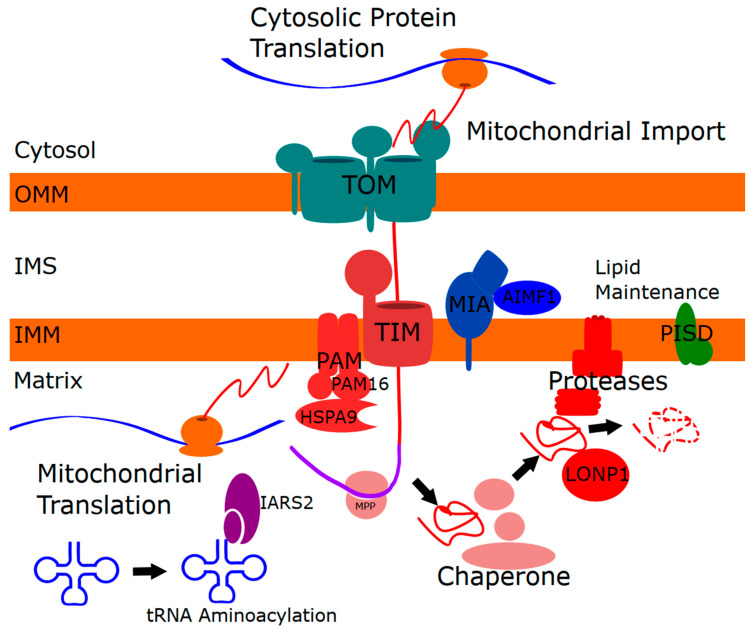
Model unifying the genes implicated in mitochondrial skeletal disorders to mitochondrial protein homeostasis. Proteases and chaperones mediate mitochondrial protein import and turnover of dysfunctional proteins. Impaired mitochondrial protein homeostasis inhibits mitochondrial protein import. IMM (inner mitochondrial membrane) lipid composition can impact proteases embedded in the IMM and is predicted to also impair mitochondrial protein import. Mitochondrial translation, which depends on tRNA aminoacylation, is essential for insertion of mtDNA-encoded proteins into the IMM, and is predicted to impact IMM proteases and/or mitochondrial protein import.

**Table 1 ijms-21-08327-t001:** Genes implicated in mitochondrial skeletal disorders, indicating formal OMIM (Online Mendelian Inheritance in Man) disease designations, gene names, protein functions, and phenotypic descriptions.

Gene	Disorder	Gene Name/Protein Function	Skeletal Phenotypes and Anomalies	Notable Reported Characteristics
*LONP1*	CODAS (cerebral, ocular, dental, auricular and skeletal): OMIM 600373	Lon peptidase 1, mitochondrial. Matrix ATP-dependent protease.	Spondylo-epi-(meta)-physeal dysplasia, short stature, facial dysmorphism, hip dysplasia	Cataracts, developmental delay, dental, hearing loss
*HSPA9*	EVEN-PLUS (epiphyseal, vertebral, ear, nose, plus associated findings): OMIM 616854	Heat shock protein family A (Hsp70) member 9 (aka Mortalin, mtHsp70, GRP75)/Mitochondrial chaperone.	Spondylo-epi-(meta)-physeal dysplasia, short stature, facial dysmorphism, scoliosis, hip dysplasia	Cataracts, cardiac malformations, dental, genital anomalies, developmental delay
*IARS2*	CAGSSS (cataracts, growth hormone deficiency, sensory neuropathy, sensorineural hearing loss, and skeletal dysplasia): OMIM 616007	Isoleucyl-tRNA synthetase 2, mitochondrial/tRNA synthetase	Spondylo-epi-(meta)-physeal dysplasia, short stature	Cataracts, neurodevelopmental delay, seizures, peripheral neuropathy hearing loss, growth hormone deficiency
*PAM16*	SMDMDM Spondylometaphyseal dysplasia, Megarbane-Dagher-Melki-type: OMIM 613320	Presequence translocase associated motor 16. ( aka MAGMAS, TIMM16)/Involved in mitochondrial protein import.	Severe spondylodysplastic dysplasia, short stature, facial dysmorphism	Cardiomyopathy, developmental delay
*AIFM1*	SEMD-HL (Spondyloepimetaphyseal dysplasia, X-linked, with hypomyelinating leukodystrophy): OMIM 300232	Apoptosis inducing factor mitochondria associated 1/Involved in mitochondrial protein import, apoptosis and assembly of mitochondrial oxidative. phosphorylation complexes.	Spondylo-epi-(meta)-physeal dysplasias, short stature, midface hypoplasia	Myelination, progressive neurodegeneration of the central and peripheral nervous system
*PISD*	SEMD (Spondylometaphyseal dysplasia). Liberfarb syndrome: OMIM 618889	Phosphatidylserine decarboxylase/Converts phosphatidylserine to phosphatidylethanolamine in the IMM.	Spondylo-epi-(meta)-physeal dysplasias, short stature, mid-face hypoplasia	Cataracts, white matter changes

**Table 2 ijms-21-08327-t002:** Genes encoding mitochondrial proteases and chaperones implicated in mitochondrial disease, indicating formal OMIM (Online Mendelian Inheritance in Man) disease designations, gene names, protein functions, and phenotypic descriptions. Disease phenotypes indicted in bold are also common in mitochondrial skeletal disorders.

Gene	Disorder	Gene Name/Protein Function	Reported Skeletal Anomalies	Notable Reported Characteristics
*CLPB*	MEGCANN (3-methylglutaconic aciduria, type VII, with cataracts, neurologic involvement and neutropenia): OMIM 616271	Caseinolytic mitochondrial matrix peptidase chaperone subunit B/Protein disaggregase associated with IMM (inner mitochondrial membrane.	Extremity rhizomelia, **impaired growth**, facial dysmorphism	**Cataracts**, neurologic deterioration, 3-methylglutaconic aciduria, neutropenia
*HSPD1*	SPG13 (Spastic paraplegia 13, autosomal dominant: OMIM 605280. Leukodystrophy, hypomyelinating, 4: OMIM 612233	Heat shock protein family D (Hsp60) member/Matrix chaperone.	N/A	**Dilated cardiomyopathy**, leukodystrophy, hypotonia, psychomotor developmental delay, spastic paraplegia
*CLPP*	PRLTS3 (Perrault syndrome 3): OMIM 614129	Caseinolytic mitochondrial matrix peptidase proteolytic subunit/Mitochondrial protease associated with IMM.	**Short stature**, facial dysmorphism	Premature ovarian failure, ataxia, microcephaly, learning difficulties, sensorineural hearing loss
*SPATA5*	EHLMRS (Epilepsy, Hearing Loss and Mental Retardation Syndrome): OMIM 613940	Spermatogenesis associated 5. AAA family of ATPases/Unclear molecular function. Role in maintaining mitochondrial function.	**Short stature**, scoliosis, hip dysplasia	**Cataracts**, epilepsy, hearing loss and intellectual disability
*HTRA2*	MGCA8 (3-methylglutaconic aciduria type VII): OMIM 617248	HtrA serine peptidase 2/IMS (inner membrane space) protease associated with IMM.	N/A	**Cataracts**, 3-methylglutaconic aciduria, seizures, hypotonia, abnormal movements, neutropenia
*AFG3L2*	OPA12 (Optic atrophy 12): OMIM 618977. SPAX5 (Spastic ataxia 5, autosomal recessive): OMIM 614487. SCA28 (Spinocerebellar ataxia 28): OMIM 610246	AFG3 like matrix AAA peptidase subunit 2/IMM protease.	N/A	Optic atrophy, spinocerebellar ataxia, spastic ataxia, chronic progressive external ophthalmoplegia
*SPG7*	SPG7 (Spastic paraplegia 7, autosomal recessive): OMIM 607259	SPG7 matrix AAA peptidase subunit, paraplegin/IMM protease.	N/A	Spastic paraplegia, ataxia, optic atrophy, cortical atrophy, cerebellar atrophy, chronic progressive external ophthalmoplegia

**Table 3 ijms-21-08327-t003:** Genes encoding mitochondrial import proteins implicated in mitochondrial disease, indicating formal OMIM (Online Mendelian Inheritance in Man) disease designations, gene names, protein functions, and phenotypic descriptions. Disease phenotypes indicted in bold are also common in mitochondrial skeletal disorders.

Gene	Disorder	Gene Name/Protein Function	Reported Skeletal Anomalies	Notable Reported Characteristics
*TOMM70*	Multi-OXPHOS deficiencies: PMID 31907385	Translocase of outer mitochondrial membrane 70/Mitochondrial import.	**Short stature**	Developmental delay, microchephaly, severe anemia, lactic acidosis
*TIMM50*	MGCA9 (3-methylglutaconic aciduria, type IX): OMIM 617698	Translocase of inner mitochondrial membrane 50/Mitochondrial import.	**Short stature**, dysmorphic facial features, hip dysplasia, scoliosis and osteoarticular issues	**Cardiomyopathy, left ventricle dilation, cardiorespiratory arrest**, 3-methylglutaconic aciduria, early-onset seizures, severely delayed psychomotor development, intellectual disability, hypotonia or spasticity
*AGK*	Senger syndrome, MTDPS10 (cardiomyopathic mitochondrial DNA depletion syndrome-10): OMIM 212350. CTRCT38 (Cataract 38, autosomal recessive): OMIM 614691	Acylglycerol kinase/Roles in mitochondrial lipid metabolism and mitochondrial import.	N/A	**Cataracts, hypertrophic cardiomyopathy**, skeletal myopathy, exercise intolerance
*DNAJC19*	DCMA (Dilated Cardiomyopathy with Ataxia), 3-methylglutaconic aciduria type V (MGCA5): OMIM 610198	DNAJ heat shock protein family (Hsp40) member C19/Roles in mitochondrial import and cardiolipin metabolism.	**Growth retardation**	**Cataracts, dilated cardiomyopathy**, ataxia, 3-methylglutaconic aciduria, genital anomalies
*GFER*	MPMCD (Myopathy, mitochondrial progressive, with congenital cataract and developmental delay): OMIM 613076	Growth factor, augmenter of liver regeneration. (aka ERV1)/Mitochondrial import MIA (mitochondrial import and assembly) component.	**Short stature**, facial dysmorphology, scoliosis, hip dysplasia	**Cataracts**, intellectual disability, hearing loss, hypotonia, developmental delay
*PMPCA*	SCAR2 (Autosomal Recessive Spinocerebellar Ataxia-2): OMIM 213200	Peptidase, mitochondrial processing subunit alpha/MTS (mitochondrial targeting sequence) cleavage.	N/A	**Cataracts**, spinocerebellar ataxia, intellectual disability
*PMPCB*	MMDS6 (Multiple mitochondrial dysfunctions syndrome 6): OMIM 617954	Peptidase, mitochondrial processing subunit beta/MTS cleavage.	N/A	Early onset severe neurodegeneration, hypotonia, intellectual disability, seizures, microcephaly, motor abnormalities
*MIPEP*	COXPD31 (Combined oxidative phosphorylation deficiency 31): OMIM 617228	Mitochondrial intermediate peptidase/MTS cleavage.	**Short stature**, facial dysmorphology	**Cataracts, left ventricular contraction, dilated cardiomyopathy**, global developmental delay, severe hypotonia, epilepsy, microcephaly
*XPNPEP3*	NPHPL1 (Nephronophthisis-like nephropathy 1): OMIM 613159	X-prolyl aminopeptidase 3/MTS cleavage. Role in cilia.	N/A	**Hypertrophic dilated cardiomyopathy**, renal failure, ciliopathy, essential tremor, hearing loss, muscle fatigue, seizures, and developmental delay
*IMMP2L*	Neurodevelopmental disorders: PMID 25478008	Inner mitochondrial membrane peptidase subunit 2/MTS cleavage.	N/A	Autism spectrum disorder, attention-deficit hyperactivity disorder, and schizophrenia

**Table 4 ijms-21-08327-t004:** Genes regulating mitochondrial lipid homeostasis implicated in mitochondrial disease, indicating formal OMIM (Online Mendelian Inheritance in Man) disease designations, gene names, protein functions, and phenotypic descriptions. Disease phenotypes indicted in bold are also common in mitochondrial skeletal disorders.

Gene	Disorder	Gene Name/Protein Function	Reported Skeletal Anomalies	Notable Reported Characteristics
*TAZ*	Barth syndrome: OMIM 302060	Tafazzin/Remodeling of cardiolipin acyl side chains.	**Pre-pubertal growth delay**, facial dysmorphism	**Dilated cardiomyopathy**, 3-methylglutaconic aciduria, neutropenia, motor delay
*SERAC1*	MEGDEL (3-methylglutaconic aciduria with deafness, encephalopathy, and Leigh-like syndrome):OMIM 614739	Serine active site containing 1/Mediates phospholipid exchange.	**Short stature**, scoliosis, dysmorphology	Impaired psychomotor function, encephalopathy, deafness, 3-methylglutaconic aciduria, spasticity
*OPA3*	Costeff Syndrome (3-methylgutaconic aciduria, type III; MGCA3): OMIM 258501. ADOAC (Autosomal Dominant Optic Atrophy and Cataract): OMIM 165300	Outer mitochondrial membrane lipid metabolism regulator OPA3/Implicated in lipid metabolism	**Growth retardation**	**Cataracts**, optic atrophy, early-onset extrapyramidal movement disorder, and cognitive deficits, 3-methylglutaconic aciduria, lipodystrophy
*AGK*	Senger syndrome, MTDPS10 (cardiomyopathic mitochondrial DNA depletion syndrome-10): OMIM 212350. CTRCT38 (Cataract 38, autosomal recessive): OMIM 614691	Acylglycerol kinase/Roles in mitochondrial lipid metabolism and mitochondrial import.	N/A	**Cataracts**, hypertrophic cardiomyopathy, skeletal myopathy, exercise intolerance
*DNAJC19*	DCMA (Dilated Cardiomyopathy with Ataxia), 3-methylglutaconic aciduria type V (MGCA5): OMIM 610198	DNAJ heat shock protein family (Hsp40) member C19/Roles in mitochondrial import and cardiolipin metabolism.	**Growth retardation**	**Cataracts, dilated cardiomyopathy**, ataxia, 3-methylglutaconic aciduria, genital anomalies

**Table 5 ijms-21-08327-t005:** Subset of genes encoding mitochondrial aminoacyl tRNAs implicated in mitochondrial disease, indicating formal OMIM (Online Mendelian Inheritance in Man) disease designations, gene names, and phenotypic descriptions. Disease phenotypes indicted in bold are also common in mitochondrial skeletal disorders.

Gene	Disorder	Gene Name	Reported Skeletal Anomalies	Notable Reported Characteristics
*LARS2*	PRLTS4 (Perrault syndrome 4): OMIM 615300	Leucyl-tRNA synthetase 2, mitochondrial.	Dysmorphic facial features, scoliosis, Marfan habitus	Premature ovarian failure, sensorineural hearing loss
*HARS2*	PRLTS 2 (Perrault syndrome 2): OMIM 614926	Histidyl-tRNA synthetase 2, mitochondrial.	N/A	Premature ovarian failure, sensorineural hearing loss
*VARS2*	COXPD20 (Combined oxidative phosphorylation deficiency 20): OMIM 615917	Valyl-tRNA synthetase 2, mitochondrial.	**Growth deficiency**, facial dymorphisms, hip dysplasia	**Hypertrophic cardiomyopathy**, muscle weakness, hypotonia, central neurologic disease, epilepsy
*WARS2*	NEMMLAS (Neurodevelopmental disorder, mitochondrial, with abnormal movements and lactic acidosis, with or without seizures): OMIM 617710	Tryptophanyl tRNA synthetase 2, mitochondrial.	**Short stature/growth retardation**, dysmorphic features	Delayed psychomotor development, intellectual disability, abnormal motor function, seizures
*GARS*	CMT2D (Charcot–Marie–Tooth disease, type 2D): OMIM 601472. HMN5A (Neuronopathy, distal hereditary motor, type VA): OMIM 600794	glycyl-tRNA synthetase 1. Mitochondrial and cytosolic.	**Growth retardation**, facial features, scoliosis	Axonal neuropathy, distal motor neuronopathy

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
