# Peer review of "Skeletal Phenotypes Due to Abnormalities in Mitochondrial Protein Homeostasis and Import"

_ijms, 2020, doi:10.3390/ijms21218327_

Round 1
Reviewer 1 Report
Your review paper is important in mitochondrial field since indeed mitochondria play a role in many diseases. You provide a wealth of information drawn from the most current knowledge of mitochondrial physiology and how many vital cellular processes depend on mitochondrial function.
However, you conflate primary mitochondrial disease (PMS) with diseases that happen to use mitochondria as one of the stages in their pathophysiologic process to cause a mechanism that is different from what is traditionally associated with PMS. If I understand correctly, you call mitochondrial disease any disease that happens to involve mitochondria (13 mtDNA genes and 1500 nuclear genes) which is incorrect. Yet you state that "it remains unclear exactly how impaired mitochondrial protein homeostasis negatively impacts skeletal development." So are you implying that it's a mere correlation rather than causation (there may be causation but state that evidence is currently unknown)?
You also mention many diseases in your tables that don't include skeletal problems. You use growth retardation as evidence of skeletal involvement but GR is a part of many diseases that don't necessarily involve bone metabolism (think of chromosomal disorders, cystic fibrosis, growth hormone deficiency, Noonan and Russell Silver syndromes, etc.). I'm surprised you haven't listed fatty acid oxidation disorders and peroxisomal disorders – they also involve mitochondria. In other words, you can’t bring more confusion that already exists among clinicians who already significantly overdiagnose mitochondrial disease. The diseases listed cannot be called mitochondrial diseases. You can say that when you know that mitochondrial process is what caused them, but you state that you don’t know that.
I suggest you entitle your paper as skeletal (or many other) disorders with mitochondrial involvement, not even dysfunction. You can’t call mito dysfunction and disease everything that emanates from mitochondria. For all constructive purposes, mitochondria could just be one of the stations where one or two of multiple processes occur that are necessary for a particular condition that is listed.
I think you can re-write the paper making it clear that mitochondrial disease does not cause skeletal anomalies. You have no evidence to make a direct link as you state so yourself. Change it to all of the important diseases that you’ve mentioned that involve mitochondria as a part of their pathophysiology but what exactly role it plays is unknown. You can speculate with all the research review that you’ve done but do not define anything that is based on pure speculation.
You cover many more abnormalities than skeletal. Your tables include genes that are implicated in cardiomyopathy, developmental delay, hypotonia, ataxia, autism, psychiatric disorders – how do these involve skeleton?
Author Response
We would like to thank the reviewer for their positive comments and thoughtful feedback on our manuscript. The reviewer makes an excellent point about mitochondrial diseases as they are classically defined: mitochondria are best known for their role in oxidative phosphorylation, and this is overall the prevailing mechanism defining the pathogenesis of mitochondrial disorders. However, as a consequence of the broad availability of genomic tools, many connections have been made in recent years between novel disease phenotypes and mitochondrial genes. We recognize this has resulted in some ambiguity around the definitions of mitochondrial disease, which in some recent reviews has expanded to include disorders beyond the classical definitions, including impairments to mitochondrial protein homeostasis (PMID: 27775730; 28415858; 29903433; 30683676; 27659608). Thus, we consider mitochondrial disease to include diseases that are caused by mitochondrial dysfunction, including, but not limited to oxidative phosphorylation. With respect to the mitochondrial skeletal disorders that we discuss, it is quite clear that the affected proteins are indeed mitochondrial, and that the pathogenic variants in these proteins impair mitochondrial function. Notably, the mitochondrial skeletal disorder genes, as well as the additional genes listed in the included tables, are listed as mitochondrial disease genes in multiple papers (PMID: 29903433; 28415858; 31021000; 32454403). Thus, we are not unilaterally calling these mitochondrial diseases. In an effort to address the reviewer’s important comment while still recognizing the broader landscape of mitochondrial dysfunction in the genomics era, we have altered the manuscript’s introduction to draw attention to this important issue.
In addition, we have made adjustments in several areas of the manuscript to reflect the reviewer’s suggestions that these are disorders of mitochondrial proteins that have a skeletal phenotype. However, we would like to argue, and have proposed in the manuscript, that there is still room for the term ‘mitochondrial skeletal disorders’. We have suggested that readers consider this new term to replace the previously used term ‘mitochondrial chaperonopathies’, as some of these disorders are not a classical chaperonopathies. Overall, while we have softened the approach to using this term throughout the manuscript, we have left it in some sections.
With respect to the additional genes, phenotypes and diseases listed in tables 2-5, these were included based on the known functions of the encoded genes, which are related to the functions of the mitochondrial skeletal disorder genes. They are meant as a comparison based on shared molecular functions. The fact that they also share several other phenotypic features, led us to suggest a common mechanistic origin based on the shared protein functions. In other words, additional pathogenic features that may also be linked to impaired mitochondrial protein homeostasis.
With respect to growth retardation, we agree as the reviewer points out that are many things that can cause growth retardation. However, this is true for many features of mitochondrial disease, there are other types of cellular dysfunction that can cause the same phenotype, this does not mean that mitochondrial dysfunction cannot be involved. We have added a note in the main text in this regard.
We are not saying that all mitochondrial diseases cause skeletal abnormalities, rather that a subset of mitochondrial diseases can cause skeletal abnormalities. This statement is completely consistent with the heterogeneous nature of mitochondrial diseases, which have a variety of phenotypes. Moreover, although we don’t know the exact molecular mechanism leading to the skeletal dysplasia, this is true for many mitochondrial disease phenotypes, and does not exclude the strong link that mitochondrial dysfunction is the underlying cause. Overall, we feel that it is important to emphasize the skeletal features can be a consequence of mitochondrial dysfunction, as this will aid in patient diagnosis.
Reviewer 2 Report
Zhao et al. have produced a comprehensive review on skeletal abnormalities in mitochondrial disease. The authors describe the consequences of impaired mitochondrial homeostasis in relation to skeletal phenotypes, such as skeletal dysplasia and metabolic bone disease. The review is well-written and contains appropriate references.
My one major comment is on the extent to which the UPRmt is described in this review. I think a more detailed description of the UPRmt is required. The authors dedicated a short paragraph at the beginning of the review; however, the response is central to several subsequent sections (e.g. mitochondrial import/homeostasis, LONP1 and CODAS, GTTP and mito tRNAs), so I would have expected a more detailed account of the signalling pathways involved in c. elegans and mammals (especially in relation to the latest studies proving the role of Atf5 in cells (Fiorese et al 2016) and in vivo models and humans (Smyrnias et al 2019, Wang et al 2019). Moreover, the authors centrally position reduced mitochondrial protein import as the reason for proteostatic stress. I believe this should also be discussed in relation to UPRmt activation and the notion that describes reduced mito protein import as a UPRmt activation signal.
Also, please address some minor typos, e.g. ATFS-1 and not ATF1 transcription factor.
Author Response
We would like to thank the reviewer for their positive comments. While we agree that the UPRmt is certainly relevant to the review, it is not the main focus here and it there are many current reviews on this topic should a reader desire to learn more about this pathway. In addition, there is currently no evidence that the UPRmt is activated in any of the disorders we discuss. While we would certainly predict the UPRmt to be activated, it still remains unclear whether this would protect or aggravate the patient phenotypes. It also notable that while ATF5 is certainly involved in the UPRmt, the evidence for its nuclear localization due to impaired mitochondrial protein import is not conclusive, as acknowledged by the authors. Thus, it remains inconclusive whether impaired protein import activates the UPRmt in mammals, as in C. elegans. Nonetheless, as suggested by the reviewer, we have added some new detail to the introductory section on UPRmt, and additional discussion in the context of mitochondrial skeletal disorders.
Round 2
Reviewer 1 Report
I would accept the paper at this stage. The authors addressed the issues that I've raised.